# Transcriptomic (DNA Microarray) and Metabolome (LC-TOF-MS) Analyses of the Liver in High-Fat Diet Mice after Intranasal Administration of GALP (Galanin-like Peptide)

**DOI:** 10.3390/ijms242115825

**Published:** 2023-10-31

**Authors:** Fumiko Takenoya, Junko Shibato, Michio Yamashita, Ai Kimura, Satoshi Hirako, Yoshihiko Chiba, Naoko Nonaka, Seiji Shioda, Randeep Rakwal

**Affiliations:** 1Department of Sport Sciences, Hoshi University School of Pharmacy and Pharmaceutical Sciences, Tokyo 142-8501, Japan; kuki@hoshi.ac.jp (F.T.); d1902@hoshi.ac.jp (M.Y.); francfranc.fragola@gmail.com (A.K.); 2Department of Functional Morphology, Shonan University of Medical Sciences, Kanagawa 244-0806, Japan; rjunko@nifty.com (J.S.); seiji.shioda@sums.ac.jp (S.S.); 3Department of Health and Nutrition, University of Human Arts and Sciences, Saitama 339-8539, Japan; satoshi_hirako@human.ac.jp; 4Laboratory of Molecular Biology and Physiology, School of Pharmacy, Hoshi University, Tokyo 142-8501, Japan; chiba@hoshi.ac.jp; 5Department of Oral Anatomy and Developmental Biology, Showa University School of Dentistry, Tokyo 142-8555, Japan; choku@dent.showa-u.ac.jp; 6Institute of Health and Sport Sciences, University of Tsukuba, Tsukuba 305-8574, Japan

**Keywords:** GALP, fat diet, liver, DNA microarray, LC-MS, metabolites, bioinformatics, obesity

## Abstract

The aim of this research was to test the efficacy and potential clinical application of intranasal administration of galanin-like peptide (GALP) as an anti-obesity treatment under the hypothesis that GALP prevents obesity in mice fed a high-fat diet (HFD). Focusing on the mechanism of regulation of lipid metabolism in peripheral tissues via the autonomic nervous system, we confirmed that, compared with a control (saline), intranasally administered GALP prevented further body weight gain in diet-induced obesity (DIO) mice with continued access to an HFD. Using an omics-based approach, we identified several genes and metabolites in the liver tissue of DIO mice that were altered by the administration of intranasal GALP. We used whole-genome DNA microarray and metabolomics analyses to determine the anti-obesity effects of intranasal GALP in DIO mice fed an HFD. Transcriptomic profiling revealed the upregulation of flavin-containing dimethylaniline monooxygenase 3 (*Fmo3*), metallothionein 1 and 2 (*Mt1* and *Mt2*, respectively), and the *Aldh1a3*, *Defa3*, and *Defa20* genes. Analysis using the DAVID tool showed that intranasal GALP enhanced gene expression related to fatty acid elongation and unsaturated fatty acid synthesis and downregulated gene expression related to lipid and cholesterol synthesis, fat absorption, bile uptake, and excretion. Metabolite analysis revealed increased levels of coenzyme Q10 and oleoylethanolamide in the liver tissue, increased levels of deoxycholic acid (DCA) and taurocholic acid (TCA) in the bile acids, increased levels of taurochenodeoxycholic acid (TCDCA), and decreased levels of ursodeoxycholic acid (UDCA). In conclusion, intranasal GALP administration alleviated weight gain in obese mice fed an HFD via mechanisms involving antioxidant, anti-inflammatory, and fatty acid metabolism effects and genetic alterations. The gene expression data are publicly available at NCBI GSE243376.

## 1. Introduction

Galanin-like peptide (GALP) is a bioactive peptide consisting of 60 amino acids. GALP was first discovered in the porcine hypothalamus, where it acts as an endogenous ligand for galanin receptors [1]. GALP-producing neurons in the hypothalamus are part of a neuronal network containing various peptides that regulate feeding behavior [2,3]. Intracerebroventricular administration of GALP has anti-obesity effects in obese mouse models [4,5,6] and reduces food intake and body weight in wild-type mice [7]. The anti-obesity effects of GALP may be due to the sympathetic nervous system-mediated suppression of appetite in peripheral tissues together with inhibition of fatty acid β-oxidation and fatty acid synthesis [8,9,10].

We hypothesized that intranasal administration would be the preferred method for translating the anti-obesity effects of GALP into clinical applications. We therefore investigated the effects of intranasal GALP in male diet-induced obese (DIO) mice. Animals were fed a high-fat diet for 14 weeks to create highly obese DIO mice. Then, GALP or saline were administered intranasally while continuing the high-fat diet (HFD) to see if GALP could suppress the hyper-obese state of the DIO mice and prevent further weight gain (Figure 1).

Furthermore, this study utilized an omics-based approach (whole-genome DNA microarray) [11,12] and metabolomics analyses (including bioinformatics analysis) to further elucidate the effects of intranasal administration of GALP on lipid metabolism in the liver tissue of DIO mice.

## 2. Results

### 2.1. Body Weight Change in DIO Mice through Intranasal GALP

The GALP-treated DIO mice did not gain weight while continuing an HFD and had significantly lower body weights than the control (saline-treated) DIO mice from Day 5 onward (Figure 2).

Similar results were reported after once-daily intranasal administration of GALP for 7 days [4], suggesting that GALP is indeed effective in preventing body weight gain.

### 2.2. Liver DNA Microarray Analysis

GALP treatment changed the expression of several genes in the livers of DIO mice. DNA microarray analysis revealed 165 genes whose expression increased (>1.5-fold difference) and 53 genes whose expression decreased (<0.75-fold difference) relative to saline treatment. The top 20 genes with altered expression are listed in Table 1.

Raw data on the total gene expressions (transcriptome) for this experiment are publicly available at the GEO (series accession number GSE243376).

### 2.3. Liver Metabolite Analysis

Metabolite analysis was performed with substances classified as acylcarnitine, fatty acids, and others. The GALP/control ratio was calculated, and the substances that increased (>1.5-fold difference) or decreased (<0.75-fold difference) following intranasal administration of GALP are shown in Table 2. Raw data are shown in Appendix A.

###  2.4. Biological Functional Enrichment Analysis

The pathway results from DAVID analysis of the DNA microarray analysis data from the GALP-treated mouse livers are shown in Table 3.

## 3. Discussion

### 3.1. DNA Microarray Analysis

#### 3.1.1. Key Genes Whose Expression Was Increased by GALP

##### Fmo3

Fmo3 is an enzyme that converts trimethylamine (TMA)—absorbed from the intestinal wall and transported to the liver—into odorless trimethylamine N-oxide (TMAO), which is then excreted in the urine. TMAO is a causative mediator of inflammatory diseases. Many studies have linked Fmo3 and TMAO to diabetes-related cardiovascular disease, while others have reported that the effects of Fmo3 are unrelated to TMA or TMAO metabolism or that Fmo3 expression and function may be related to acetaminophen-induced effects. Fmo3 reportedly has a role in protecting the liver from toxicity [13]. Although TMAO is a possible pro-inflammatory mediator, Fmo3 may play a role in metabolic diseases regardless of TMAO formation. Some reports indicate that Fmo3 is significantly involved in liver detoxification, as the loss of Fmo3 increases ammonia in the liver [14]. Furthermore, it is well known that physical activity has a positive effect on gut microbiota, and it is clear that TMAO levels increase after exercise and sports [15]. Although these findings suggest that Fmo3 may be involved in antioxidant and metabolic processes, the link between increased Fmo3 and poor health has not yet been fully elucidated. Many recent studies have reported increased Fmo3 expression in mouse models of longevity [16]. Although the relationship between aging and Fmo3 is not well understood, 40% caloric restriction, considered one of the most powerful interventions to slow the aging process, has been found to significantly upregulate Fmo3 gene expression. Overexpression of Fmo3 is associated with inflammation, metabolism, oxidative stress, and liver function and improves key liver parameters related to hepatocyte aging [17]. There are also reports that FMO activation may be a conserved mechanism to enhance protein homeostasis, improve health, extend the lifespan, etc. [18]. These factors raise many questions regarding the increased expression of Fmo3, but it is possible that GALP administration mimics the caloric restriction state or may be involved in the enhanced energy expenditure.

##### Lyve1

Recent studies have shown that obesity impairs lymphatic function in both mice and humans. Obese mice have significantly less Lyve1, a marker of lymphangiogenesis, than their controls [19]. Diet-induced obesity has been shown to significantly impair the lymphatic system, as reflected by decreased lymph flow, altered lymph node structure, and increased inflammatory tendencies [20]. Lymphatic vessels are essential for maintaining tissue fluid balance, facilitating the transport of immune cells from the periphery to the lymph nodes, and absorbing lipoproteins from the gut and tissue fat cells. Mouse models of dyslipidemia lacking apolipoprotein E (apoE^−/−^) show reduced lymphatic integrity and transport function that worsens as hypercholesterolemia progresses, suggesting that the lymphatics themselves are dysfunctional. Increased expression of the Lyve1 gene associated with GALP administration may indicate restoration of lymphatic function that has been impaired by obesity.

##### Mt1, Mt2

Metallothionein (MT) is involved in zinc and copper homeostasis, detoxification of heavy metals, and protection against cellular damage caused by oxidative, inflammatory, and stress responses. There are reports that MT can prevent the onset of obesity caused by a high-fat diet in female mice [21]. Mt1- and Mt2-null mice showed obesity and hyperleptimemia due to increased liver lipids [22], while obesity due to a high-fat diet may occur at an earlier age in MT gene knockout mice [23]. On the other hand, the MT liver content increases significantly in fasted mice [24,25].

##### Aldh1a3

Aldh1a3 is an NAD-dependent aldehyde dehydrogenase that catalyzes the formation of retinoic acid. Retinoic acid administration improves non-alcoholic fatty liver disease in high-fat diet-induced obese mice and has a therapeutic effect on hepatic lipid accumulation by increasing energy expenditure [26]. Furthermore, Aldh1a3 plays a major role in alcohol metabolism and the detoxification of aldehydes produced by lipid peroxidation. Decreased expression of Aldh1a3 correlates with hepatic lipidosis and liver fibrosis as well as increased microvascular lipidosis caused by lipid peroxidation, possibly due to an accompanying decrease in β-oxidation [27].

##### Fgfr1

Fgfr1 is a receptor for fibroblast growth factor 21 (FGF21), a regulator of lipid and glucose metabolism. It has been found to reduce fatty liver disease and steatohepatitis, a disorder of lipid metabolism in non-alcoholic fatty liver disease mice, through upward regulation of the FGF21/FGFR1 pathway [28].

#### 3.1.2. Key Genes Whose Expression Was Decreased by GALP

##### Antimicrobial Peptides: Defa3, Defa20, Defa6, Defa1, and Defb19

Defensins control the intestinal microbiota by eliminating pathogenic and opportunistic bacteria that are detrimental to the host, with little bactericidal activity against *Bifidobacterium* and *Lactobacillus* that act favorably. A high-fat diet causes changes in the composition of the intestinal microbiota; the ratio of Firmictes:Bacteroides (F/B ratio) is increased, while endotoxemia and liver inflammation occur due to barrier damage and increased levels of the cytotoxin lipopolysaccharide (LPS) [29,30]. In patients with liver disease, the antimicrobial peptide CRAMP is elevated. This peptide acts as a protective factor during liver injury in mice [31]. Similarly, the expression and secretory levels of the liver-expressed antimicrobial peptide LEAP2 are increased in mice with diet-induced lipidosis [32]. Thus, defensins are induced by pathogens and inflammatory stimuli. The observed decrease in defensin expression in the liver during GALP administration could thus be due to the amelioration of HFD-induced disruption of the gut microbiota and reduced hepatic inflammation.

##### Anti-Inflammatory Proteins: Gkn2, Tff1, and Clca1

Heterodimer formation of GKN2 and TFF1 has substantial antiproliferative and apoptosis-promoting effects, suggesting that co-secretion of TFF1 and GKN2 inhibits gastric carcinogenesis [33] and may also protect gastric mucosa from oxidative stress-induced damage [34]. Thus, as with antimicrobial peptides, the observed decrease in Gkn2 and Tff1 expression following GALP administration may be due to the amelioration of HFD-induced inflammation. Clca1 is thought to function as a mucoprotein that protects the intestinal mucus barrier. It may also be involved in the regulation of tissue inflammation during innate immune responses through the regulation of cytokine and chemokine production [35].

##### Steroid Biosynthesis Pathway: Hsd3b4, Hsd3b5, and Hsd3b1

Hsd3b4, Hsd3b5, and Hsd3b1 are all essential enzymes for steroid hormone biosynthesis. The results of the DAVID analysis showed that the top gene category suppressed by GALP treatment was cortisol synthesis and secretion. Hsd3b is involved in the synthesis of cortisol, one of the adrenal corticosteroid hormones. Cortisol is involved in liver metabolism, such as gluconeogenesis and the breakdown of fats in adipose tissue, and it has anti-inflammatory effects.

##### Bile Acid Secretion and Transport: Slco1a1/Oatp1

Slco1a1 is a transporter involved in the reuptake of bile acids into the liver [36]. Bile acids synthesized in the liver undergo enterohepatic circulation, and hepatic bile acid levels are highly dependent on the function of bile acid uptake and efflux transporters in the liver.

##### Lipid and Cholesterol Metabolism: Cyp26a1, Pnpla5, Prap1, and Fdps

Cyp26a1, a retinoic acid-metabolizing enzyme, is downregulated by glucagon and cortisol secreted during fasting. Pnpla5 is involved in diacylglycerol production and promotes lipid droplet formation [37]. It is highly activated in the liver of obese (ob/ob) mice, and its expression is suppressed by fasting [38]. Deficiency of the Prap1 gene reduces hypertriglyceridemia, weight gain, and hepatic lipidosis in mice on an HFD [39]. Fdps, a cholesterol synthesis-related gene, encodes an enzyme that catalyzes the formation of farnesyl pyrophosphate, which is an important intermediate in the biosynthesis of cholesterol and sterols.

### 3.2. Metabolite Analysis

#### 3.2.1. Metabolites Increased by GALP

##### Coenzyme Q10

Coenzyme Q10 is an endogenous lipophilic substance with redox activity incorporated into the mitochondrial respiratory chain. It is recognized as a major regenerative antioxidant that functions as an electron carrier to generate cellular energy, and it plays an essential role against oxidative damage. One of the main functions of coenzyme Q10 is as an antioxidant, as shown by reductions in oxidative stress and diet-induced inflammation in the livers of obese mice [40]. Other effects include the prevention of bile stasis and improvement of the redox environment of the liver [41]. Furthermore, in diabetic and obese KKAy mice assigned reduced coenzyme Q10 intake, the expression of a group of genes involved in fatty acid and cholesterol synthesis was decreased. This indicates that reduced coenzyme Q10 can inhibit fatty acid synthesis and activate Pparα to promote beta oxidation of fatty acids in the liver [42].

##### Hydroxyprogesterone Caproate

Progesterone supplementation with progesterone derivatives such as 17-OHPC has long been used to decrease the risk of recurrent preterm labor. Anti-inflammatory effects and blood pressure reduction due to vasodilatation are considered to be the main actions [43].

##### Oleoylethanolamide

Oleoylethanolamide (OEA) is an N-acylethanolamine-containing oleic acid, a monounsaturated fatty acid, which exerts weight loss effects by suppressing appetite and stimulating lipolysis via PPARα [44,45,46,47,48]. The appetite-suppressive effects of OEA may be produced through a vagally mediated mechanism [49]. In addition, there are reports of antioxidant and anti-inflammatory effects of OEA, such as OEA improving oxidative stress in the liver [50,51,52].

##### Erucic Acid

Erucic acid significantly decreases the expression of adipocyte marker genes and accelerates osteoblast marker genes via the suppression of PPARγ transcriptional activity [53]. Reports have suggested that yellow mustard oil rich in erucic acid can improve obesity-induced metabolic disorders [54].

##### Linolenic Acid (Alpha-Linolenic Acid)

Vegetable oils rich in α-linolenic acid exert anti-obesity effects by inhibiting lipogenesis and the upward regulation of fatty acid β-oxidation via PPARα [55,56]. For example, the “Rosa mosqueta” oil, which contains high concentrations of α-linolenic acid, has been shown in male C57BL/6J mice to reduce PPARα activation. It also prevents HFD-induced oxidative stress and inflammation and reduces fatty liver disease [57]. Meanwhile, 2-arachidonoylglycerol (2-AG) restores impaired glucose uptake and insulin resistance caused by exposure to inflammatory stress and lipid overload [58]. Many reports implicate 2-AG in neuronal, immune, metabolic, vascular, and reproductive functions, as well as in the homeostatic and hedonic aspects of feeding. Indeed, 2-AG is a natural ligand for CB1 receptors, which are involved in the regulation of food intake and energy metabolism and may stimulate hunger and food intake and modulate food preferences [59].

##### Bile Acids: Deoxycholic Acid, Taurocholic Acid, and Taurochenodeoxycholic Acid

GALP administration increased the bile acids of deoxycholic acid (DCA), taurocholic acid (TCA), and taurochenodeoxycholic acid (TCDCA). DCA is a secondary bile acid formed by intestinal bacteria from cholic acid; TCA and TCDCA are primary bile acids formed from cholic acid and kenodeoxycholic acid conjugated with taurine and have been used as cholecystokinin agents; and TCA and DCA have been reported to improve obesity via the G protein-coupled bile acid receptor Gpbar1 (TGR5)-dependent pathway [60,61]. TCDCA also acts as a signaling molecule that exerts anti-inflammatory and immunomodulatory functions via the cAMP-PKA-CREB signaling pathway induced by the TGR5 receptor [62,63].

##### 5α-Cholestan-3-One-2 and 5α-Cholestanone

Cholesterol processing involves the production of coprostanone via 4-cholestan-3-one and 5α-cholestanone as intermediate products, converting them to coprostanol as the final product. This processing, which is attributed to the intestinal microbiota, leads to direct excretion of cholesterol products in the feces, decreased cholesterol absorption, and decreased physiological total cholesterol concentration.

#### 3.2.2. Metabolites Reduced by GALP

Many of the metabolites reduced by GALP administration have antioxidant and anti-inflammatory effects.

##### Flavanone

Citrus fruits are rich in flavanones, and flavanones such as naringenin, eriodictyol, and hesperetin exhibit anti-inflammatory and antioxidant effects. Naringenin has the physiological effect of increasing fatty acid oxidation in the liver through upregulation of the gene expression of enzymes involved in peroxisomal beta oxidation in mice [64].

##### Riboflavin

Riboflavin, a water-soluble B2 vitamin, has antioxidant properties and is a central component of flavin adenine dinucleotide (FAD). FAD is a component of many enzymes requiring electron transfer and acts as a cofactor for oxidoreductase enzymes involved in carbohydrate, protein, and fat metabolism and energy production. Antioxidants (vitamin E) are found in fatty liver medications. Vitamin E prevents oxidation of fats in the body, improving the fatty liver state and reducing disease progression. Zeaxanthin is a polyunsaturated fatty acid carotenoid and an antioxidant.

##### Palmitoylethanolamide-2

Palmitoylethanolamide (PEA) is an endocannabinoid-like lipid mediator with widely demonstrated anti-inflammatory, analgesic, antibacterial, immunomodulatory, and neuroprotective effects [65].

##### Myristic Acid and Arachidic Acid

Myristic acid is used as an emulsifier, and arachidic acid is used as a surfactant. A 4 week intervention combining exercise training and dietary restriction in obese subjects was associated with a decrease in the serum fatty acids of myristic acid, stearic acid, arachidic acid, behenic acid, palmitoleic acid, and dihomo-γ-linolenic acid, as well as a decrease in body fat [66].

##### Ursodeoxycholic Acid (UDCA)

Among the reduced metabolities was the secondary bile acid ursodeoxycholic acid (UDCA), which has diuretic, choleretic, anti-inflammatory, digestive, and hepatic ameliorative properties. UCDA is approved by the US Food and Drug Administration for the treatment of primary cholestatic cholangitis. It has also been reported to reduce metabolic dysfunction in mice with HFD-induced obesity [67].

### 3.3. Pathway Analysis of Expression Dissimilarity Genes

#### 3.3.1. KEGG Upregulated

##### Fatty Acid Elongation (mmu00062) and the Biosynthesis of Unsaturated Fatty Acids (mmu01040)

The genes ACOT1, ACOT3, and ELOVL7 were categorized in the pathway of fatty acid elongation and the synthesis of unsaturated fatty acids. These genes are involved in fatty acid degradation through β-oxidation and fatty acid elongation and lead to an increase in the percentage of unsaturated fatty acids. Unsaturated fatty acids have generally been found to prevent atherosclerosis and blood clots, lower blood pressure, and reduce LDL cholesterol. Our metabolite analysis showed a high increase in unsaturated fatty acids after GALP administration.

##### Retinol Metabolism (mmu00830)

Retinol is a known regulator of adipocyte differentiation.

##### Melanoma (mmu05218)

The genes classified here included Gadd45b and FGFR1. Gadd45b is thought to be involved in a variety of responses to cell injury, including cell cycle checkpoints, apoptosis, and DNA repair. Recently, evidence has accumulated that Gadd45b deficiency promotes premature aging and skin senescence [68], and FGFR1 has been reported to ameliorate impaired lipid metabolism in mice with non-alcoholic fatty liver disease through upward regulation of the FGF21/FGFR1 pathway [69].

#### 3.3.2. KEGG Downregulated

##### Staphylococcus aureus Infection (mmu05150) and Salmonella Infection (mmu05132)

Obese children reportedly have fewer bifidobacteria and more *Staphylococcus aureus* in infancy than normal-weight children of the same age [70]. GALP administration appeared to reduce the number of bacteria (*Escherichia coli*, *Salmonella*, *S. aureus*, etc.) that produce toxic substances in the intestines.

##### Cortisol Synthesis and Secretion (mmu04927)

Cortisol (corticosterone) in mouse hair has been found to be higher in high-fat diet mice than in normal diet mice [71]. Elevated cortisol has been linked to overeating and metabolic syndrome, and it has also been linked to hypertension, high cholesterol, and insulin resistance.

##### Cushing’s Syndrome (mmu04934)

Cushing’s syndrome is a disease characterized by excessive cortisol secretion and characteristic physical findings. The diagnostic features of metabolic syndrome and Cushing’s syndrome are similar, and it has been proposed that excess cortisol contributes to the development of both conditions.

##### Cholesterol Metabolism (mmu04979)

Decreased cholesterol metabolism leads to obesity reduction. The genes in this category were STAR, SORT1, and PCSK9. Steroidogenic acute regulatory protein (StAR) is a transport protein that moves cholesterol from the mitochondrial outer membrane to the inner membrane and is the rate-limiting step in steroid hormone synthesis. SORT1 has been identified as a gene associated with blood LDL cholesterol levels and myocardial infarction risk, and its role in lipid metabolism has been investigated. SORT1 knockout mice fed a high-cholesterol diet showed decreased free cholesterol accumulation in the liver, increased bile acid synthesis, decreased cholesterol secretion in bile, and no gallstone formation [72]. PCSK9 inhibitors are effective treatments for hypercholesterolemia. They prevent degradation of low-density lipoprotein receptors (LDLRs), thereby promoting the uptake of LDL cholesterol into hepatocytes and lowering the level of LDL cholesterol in the blood [73].

##### Bile Secretion (mmu04976): Slco1a1, Sult2a8, and Slc22a7

Bile acid homeostasis plays a direct role in maintaining energy balance and is tightly regulated through the synthesis, reabsorption, and removal of bile acids. The transporter Slco1a1/Oatp1 transports bile acids, thyroid hormones, and various drugs into hepatocytes. SLC22A7/OAT2 may function as a glutamate efflux transporter in hepatocytes and may play an active role in the release of glutamate from the liver into the bloodstream [74]. Like Slco1a1, SLC22A7/OAT2 is involved in the transport of bile acids and drugs into hepatocytes. Recent physiological studies have identified SULT2A8 as the major bile acid-sulfonating enzyme in mice [75], while 7-OH sulfonation of bile acids with SULT2A8 is a pathway for bile acid detoxification, increasing the water solubility of bile acids, limiting enterohepatic recirculation, and promoting fecal excretion (thereby nullifying bile stasis).

### 3.4. Integrated Data Analysis

This study demonstrated that, compared with the control (saline), intranasal administration of GALP had an inhibitory effect on weight gain in obese mice consistently fed a high-fat diet. The observed effects of GALP on feeding may vary according to the route of administration, time of measurement, and animal model. For example, intracerebroventricular administration of GALP to rats results in a large increase in food intake over a short period of 1–2 h [76] and further increases high-fat food intake in DIO rats [77]. However, 24 h after administration, an anti-obesity effect occurs, reducing the food intake and body weight [78]. Conversely, experiments in mice do not show the orexigenic effects seen in rats, and several studies have reported a decrease in food intake and body weight 24 h after intracerebroventricular GALP administration [79,80,81]. Experiments in obese mice have also demonstrated the anti-obesity effects of GALP, as continuous intracerebroventricular administration to ob/ob mice for 14 days results in a sustained decrease in food intake and body weight [82]. These results suggest that the short-term feeding-promoting effect of GALP is rat-specific and that GALP as a whole is an anti-obesity peptide.

The anti-obesity effect of GALP may be due to an increase in energy metabolism, although the detailed mechanism has not yet been elucidated. We have documented an increase in core body temperature and oxygen consumption over an 8 h period following intracerebroventricular administration of GALP in rats [6]. Notably, this increase in body temperature exceeds that induced by capsaicin administration. In addition, astrocytes, a subtype of glial cells within the central nervous system, are activated by GALP [6]. Our analysis of the gene expression levels following GALP administration suggests that GALP-releasing neurons induce expression of the Cox-2 gene in astrocytes, thereby facilitating thermogenesis [6]. GALP has also been shown to upregulate gene and protein expression of the uncoupling protein UCP-1, a specific marker of brown adipose tissue known for its involvement in sympathetically driven thermogenesis. This finding provides evidence that GALP enhances energy metabolism by stimulating peripheral sympathetic nerves. Furthermore, intracerebroventricular administration of GALP induces an increase in fatty acid β-oxidation in the liver and promotes lipolysis in adipose tissue. Thus, the anti-obesity effects of GALP result from improvements in peripheral tissue lipid metabolism via the sympathetic nervous system [9].

The anti-obesity effects of GALP have been established at both the central and peripheral levels, with promising clinical implications. However, previous studies have used either intracerebroventricular or intraperitoneal administration, which are considered impractical for clinical use due to associated pain and safety concerns. We therefore conducted a comprehensive investigation into the anti-obesity potential of GALP through intranasal administration.

Previously, we investigated the brain transfer kinetics of GALP and successfully developed an efficient nasal spray delivery technique. To elucidate the efficacy of intranasal administration, we compared the transfer rates of radioactively iodinated GALP via intravenous, intracerebroventricular, and intranasal routes and found that intranasal administration had a significantly higher brain transfer rate. Indeed, the brain uptake rate of GALP was 3–5 times higher with intranasal than with intravenous administration. There was substantial uptake of GALP in the hypothalamus and hippocampus following intranasal administration, with minimal to no detectable uptake of GALP in the spleen or lymph nodes [8].

We then administered a nasal spray of GALP to wild-type ob/ob and DIO mice. Although no significant changes were observed in the wild-type mice, the two obesity models showed significant reductions in body weight and blood glucose levels [4]. No alterations in locomotor activity or taste aversion were observed, suggesting that the anti-obesity effect of GALP may be due to increased energy metabolism. In support of this, we found that the respiratory quotient was reduced by intranasal GALP. By observing the gene expression of SREBP-1 and FAS, which are involved in fatty acid synthesis in the liver, we confirmed that fatty acid synthesis was suppressed [10]. Research is being conducted to explore novel approaches for the treatment and management of obesity using peptides. Several studies have demonstrated anti-obesity effects following intranasal administration of peptides in both humans and animals. However, few have investigated the effects of peptide intranasal administration using array analysis. Moreover, these studies have not yet reached the practical implementation stage, and clinical application may require a considerable amount of time.

Ours is probably the first study where gene expression and a metabolomics approach have been used to investigate the effects of intranasal GALP administration on the mechanisms of feeding inhibition and weight loss. Feeding inhibition effects may involve an increase in the postprandial satisfaction inducer OEA and a decrease in the expression of genes involved in the synthesis pathway of cortisol, which has an appetite-promoting effect. Weight loss mechanisms may involve decreases in cholesterol and fatty acids, a decrease in cholesterol and adipogenesis gene expression, improvement in fatty acid composition, and changes in antibacterial and anti-inflammatory peptides as well as intestinal microflora (Figure 3).

#### 3.4.1. Feeding Inhibition Mechanism

##### OEA

Omics (gene and metabolite) analyses of the liver after intranasal administration of GALP suggest that OEA and the cortisol synthesis pathway may be involved in appetite suppression. OEA is a type of endogenous N-acylethanolamine that acts as a signal from the gut to the brain and is a target for new treatments for obesity and eating disorders. The inhibition of eating brought about by OEA may result from a combination of humoral pathways, afferent vagal pathways via activation of PPARα, and others [44,83,84,85,86].

##### Cortisol Synthesis-Related Genes

A decrease in cortisol synthesis-related genes was identified by the KEGG pathway as being involved in feeding inhibition. Glucocorticoids stimulate food intake [87], and elevated glucocorticoid levels are associated with abdominal fat accumulation [88]. Furthermore, clinical symptoms characterized by high cortisol levels, such as those seen in patients with Cushing’s syndrome and Prader–Willi syndrome, are associated with abdominal obesity [89]. Increased hepatic accumulation of lipids by glucocorticoids is mediated by multiple mechanisms, including increased food intake, stimulation of glycogenesis, stimulation of fatty acid uptake, inhibition of β-oxidation of fatty acids, and increased release of fatty acids from adipose tissue [90,91]. As described above, both OEA and cortisol are likely to be important in their effects on weight loss, as there are many reports of their involvement in lipid metabolism, including beta-oxidation of fatty acids, as well as feeding inhibition.

#### 3.4.2. Weight Loss Mechanism

##### Decreased Expression of Lipid Synthesis-Related Genes

DNA microarray analysis revealed that GALP treatment decreased the expression of many lipid synthesis-related genes (lipid and cholesterol synthesis genes (Fdps, Gpam, Mvd, Sc5d, Insig2, Insig1, and Pcsk9), fatty acid synthesis genes (Elovl3, Fasn, and Me1), fat absorption-promoting gene (Prap1), and fat droplet-forming gene (Pnpla5)). PCSK9 inhibitors are particularly effective in the treatment of hypercholesterolemia, lowering blood LDL cholesterol levels by inhibiting the binding of PKSK9 to the low-density lipoprotein receptor (LDLR) and promoting the uptake of LDL cholesterol into liver cells [73]. GALP administration suppressed PCSK9 gene expression, which may be one of the mechanisms of weight loss.

##### Improved Fatty Acid Composition

Unsaturated fatty acids improve insulin efficacy and decrease liver fat mass. Therefore, “switching from saturated to unsaturated fatty acids” is considered important in the treatment of fatty liver disease. Our metabolite analysis revealed that GALP administration predominantly decreased saturated fatty acids and increased unsaturated fatty acids. In addition, pathway analysis showed increases in fatty acid elongation (mmu00062) and biosynthesis of unsaturated fatty acids (mmu01040), both of which promote fatty acid unsaturation. Thus, GALP administration promotes the switch from saturated fatty acids to unsaturated fatty acids.

##### Antibacterial and Anti-Inflammatory Peptides and Intestinal Microflora

Excessive intake of a high-fat diet reduces the intestinal barrier function and increases the permeability of the intestinal mucosa, resulting in systemic inflammation and intestinal microbiota abnormalities due to foreign body invasion. Intake of a diet high in fat and calories leads to bulimia and obesity, accompanied by chronic mild systemic inflammation and changes in the intestinal microbiota [92]. Defensins production is induced by infectious stimuli but also by inflammation [93]. Recently, the chronic inflammatory inhibitory effect of defensins has been proposed for the treatment of diabetic wounds [94], and α-defensins have been found to induce a favorable blood metabolic profile, increased systemic lipolysis, and decreased liver fat accumulation [95].

A high-fat diet promotes LPS-induced intestinal barrier dysfunction, and serum LPS is now accepted as a surrogate marker for assessing in vivo intestinal permeability [96]. GALP administration induces *S. aureus* infection (mmu05150) and *Salmonella* infection (mmu05132) pathways, while the antimicrobial peptide defensin expression is highly suppressed. These results suggest that GALP treatment repairs the intestinal microbiota and prevents the movement of inflammatory agents such as LPS, which are associated with the growth of gram-negative bacteria, from the intestinal tract to the liver. Furthermore, the anti-inflammatory effects of OEA treatment have been suggested to shift the composition of the gut microbiota to a “lean-like phenotype” [97,98]. It is also likely that OEA is responsible for the changes in gut microbiota composition induced by GALP administration.

##### Bile Uptake and Excretion

Reabsorption and excretion of bile acids are known to be enhanced by UDCA administration (i.e., UDCA promotes enterohepatic circulation of bile acids) [99]. In addition to a decrease in the expression of bile uptake and excretion-related transporter genes involved in enterohepatic circulation, our results confirm that a reduction in UDCA decreases Slco1a1/OATP1 gene expression. It has long been known that dietary fat stimulates both the synthesis of bile acids and the efflux of bile through the gallbladder, and several studies have shown a direct correlation between fat intake and bile acid secretion and synthesis [100]. In light of these considerations, the possible reasons for the observed decreases in bile uptake- and excretion-related transporter gene expression and UDCA in the present study are as follows: (1) suppression of lipid and cholesterol synthesis-related gene expression by GALP may result in a decrease in the required bile acids, and (2) HFD-induced mild inflammation of the intestinal tract and improved intestinal microflora composition may reduce the need to promote fecal excretion of cytotoxic bile acids and other substances.

##### β-Oxidation of Fatty Acids

Changes in the expression of genes directly involved in the β-oxidation of fatty acids were not evident from the results of our study. However, coenzyme Q10 [101], OEA [102], and linolenic acid [103] are known to be peroxisome proliferator-activated receptor (PPAR) activators. Furthermore, the FMO3 knockdown mice showed significantly decreased expression of genes involved in glycogenesis and lipogenesis, as well as decreased levels of unsaturated fatty acids and expression of PPARα target genes in the liver. Thus, FMO3 may modulate the catabolism of unsaturated fatty acids and indirectly influence PPARα activation [104]. PPARs primarily regulate the expression of gene networks involved in adipogenesis, lipid metabolism, inflammation, and metabolic homeostasis. They are activated by dietary fatty acids and their metabolites and thus act as lipid sensors that, when activated, can significantly redirect metabolism. PPARα and PPARγ play important roles in fatty acid catabolism and storage, respectively, and PPARα activation induces liver PPARα/γ. Therefore, intranasal administration of GALP may enhance fatty acid β-oxidation via PPARα activation.

## 4. Materials and Methods

### 4.1. Preparation of the DIO Mice

To observe changes in body weight induced by intranasal GALP administration, the C57BL/6 mice were fed a high-fat diet ad libitum for 14 weeks to create DIO mice (Figure 1). The DIO mice used for GALP administration weighed at least 40 g. The mice continued to have ad libitum access to an HFD throughout the administration of GALP or saline.

All experimental procedures involving animals were approved by the Institutional Animal Care and Use Committee of Hoshi University. The mice were bred and maintained under specific pathogen-free conditions in the animal facility of Hoshi University.

### 4.2. Intranasal Administration of GALP to the DIO Mice

Sixteen DIO mice were divided into two groups, GALP administration and control (saline) (*n* = 8 per group), and acclimatized for 1 week with once-daily intranasal administration of 2 µL of saline. Intranasal administration was performed by pushing a plastic 10 μL gel-loading pipette tip attached to a 10 μL micropipette through the naris.

Porcine GALP (1–60) purchased from Bachem AG (Bubendorf, Switzerland) was dissolved in saline containing 5% α-cyclodextrin to a concentration of 1 nmol. After acclimatization, the mice received once-daily intranasal administration of GALP or saline for 11 consecutive days. The body weights were measured at the time of each intranasal administration.

Statistical analyses were performed using GraphPad Prism 7 (GraphPad Software, Boston, MA, USA). A Student’s *t* test was used to compare between the two groups. Data are presented as median ± standard deviation (SD).

### 4.3. Dissection and Sampling

After completion of the GALP or saline treatment, the animals were sacrificed by decapitation. The livers were sampled, immediately powdered in liquid nitrogen, and stored at −80 °C until total RNA extraction and metabolome analyses.

### 4.4. DNA Microarray Analysis

Total RNA was extracted from the powdered mouse liver tissue using an RNeasy Mini Kit (74104, Qiagen). To verify the quality of this RNA, the yield and purity were determined spectrophotometrically with DS-11 (DeNovix, Wilmington, DE, USA) and confirmed using formaldehyde-agarose gel electrophoresis. To check the quality of the synthesized cDNA using an Affinity Script QPCR cDNA synthesis kit (600559, Agilent), a PCR reaction was performed to confirm the expression of house-keeping genes (beta-actin or GAPDH) using Emerald Amp PCR Master (RR300A, Takara, Japan). The PCR products were separated on a 1.5% agarose gel and visualized with ethidium bromide staining under UV light.

The total RNA extracted from the liver tissue of the control mice (*n* = 8) was pooled, and 500 ng were used for DNA microarray analysis. The total RNA extracted from the liver tissue of the GALP-treated mice was used for DNA microarray analysis in two groups: those with greater weight loss (*n* = 4) and those with less weight loss (*n* = 4) (Whole Mouse Genome DNA Microarray 4x44K, G2514F). The microarray experiments were performed as described previously [11,12]. Briefly, to select differentially expressed genes with the dye-swap approach, the total RNA (500 ng) was labeled with either Cy3 or Cy5 dye using an Agilent Low RNA Input Fluorescent Linear Amplification Kit (Agilent Technologies Inc., Santa Clara, CA, USA). Fluorescently labeled targets of the control and GALP-treated samples were hybridized to the same microarray slide with 60 mer probes. A flip labeling (dye-swap or reverse labeling with Cy3 and Cy5 dyes) procedure was followed to nullify the dye bias associated with unequal incorporation of the two Cy dyes into cDNA. Hybridization and washing processes were performed according to the manufacturer’s instructions, and the hybridized microarrays were scanned using an Agilent Microarray G2505C scanner.

Initially, in the process of identifying the genes involved in GALP-induced weight loss in the microarray analysis, we thought it would be effective to group the genes according to the degree of weight loss. However, as there was no significant difference between the results of each group, we decided to proceed with the analysis by selecting the genes that showed common expression changes in each group, as shown below. To detect significantly differentially expressed genes between the control (*n* = 8) and GALP-treated (greater weight loss; *n* = 4) groups and between the control and GALP-treated (less weight loss; *n* = 4) groups, each slide image was processed with Agilent Feature Extraction software (version 9.5.3.1). This program measures the Cy3 and Cy5 signal intensities of whole probes. As dye-bias tends to be dependent on the signal intensity, the software selected probes using a set with a rank consistency filter for dye normalization. Said normalization was performed with locally weighted linear regression (LOWESS), which calculates the log ratio of dye-normalized Cy3 and Cy5 signals as well as the final error of the log ratio. The significance (P) value is based on the propagation error and universal error models. In this analysis, the threshold of significant differentially expressed genes was <0.01 (for the confidence that the feature was not differentially expressed). Erroneous data generated due to artifacts were eliminated before data analysis using the software.

The Functional_Categories (KEYWORDS) and pathway (KEGG pathway) of the list of variable genes selected following DNA microarray analysis were analyzed using Database for Annotation, Visualization and Integrated Discovery (DAVID) v6.8.

The obtained whole-genome DNA microarray data of the GALP experiment was submitted to NCBI’s GeneExpression Omnibus (https://www.ncbi.nlm.nih.gov/geo/query/acc.cgi?acc=GSE243376; accessed on 30 October 2023) and is available under the GEO with series accession number GSE243376.

### 4.5. Mouse Liver Metabolome Analysis by LC-TOFMS

The combined liver powder samples from each treatment group of eight mice were further ground in liquid nitrogen before use for downstream liquid chromatography/time-of-flight/mass spectrometry (LC-TOFMS) pretreatment.

The pooled liver powder was weighed (control: 32.9 mg; GALP: 31.7 mg), and 500 μL of 1% formic acid-acetonitrile solution (internal standard concentration: 10 μL) was added. The mixture was crushed once at 1500 rpm for 120 s using a crushing machine under cooling, 167 μL of Milli-Q water was added, and more crushing was performed at 1500 rpm for 120 s. After centrifugation (2300× *g*, 4 °C, 5 min), the supernatant was collected. To the precipitate, 500 μL of 1% formic acid-acetonitrile solution and 167 μL of Milli-Q water were added and stirred. After a second centrifugation (2300× *g*, 4 °C, 5 min), the supernatant was collected again and mixed with the previous supernatant. This combined supernatant was transferred to two 300 µL × 300 µL tubes (Nanosep 3K Omega, Pall Corporation, Port Washington, NY, USA) and centrifuged (9100× *g*, 4 °C, 120 min) for marginal filtration treatment. Phospholipids were removed using solid phase extraction, and the filtrate was dried, dissolved in 200 μL of 50% isopropanol solution (*v*/*v*), and subjected to LC-TOFMS measurement.

For both the cationic metabolites (positive mode) and anionic metabolites (negative mode), the LC system was an Agilent 1200 series RRLC system SL (Agilent Technologies Inc., Santa Clara, CA, USA), and the MS system was an AgilentLC/MSD TOF (Agilent Technologies). The analytical conditions were as follows: the column was an ODS column (2 × 50 cm, 2 μm), mobile phase A was H_2_O/0.1% HCOOH, mobile phase B was isopropanol: acetonitrile: H_2_O (65:30:5)/0.1% HCOOH, 2 mM HCOONH4, the gradient conditions were 0–0.5 min = B 1%, 0.5–13.5 min = B 1–100%, and 13.5–20 min = B 100%, and the flow rate was 0.3 mL/min. The detected peaks (57 cation and 47 anion) were searched against all substances registered in the HMT Metabolite Library and the Known-Unknown Library based on the *m*/*z* and MT values.

## 5. Conclusions

This controlled study examined the effects of intranasal GALP administration in obese mice with continuous access to a high-fat diet. We found that the expression of genes related to lipid and cholesterol synthesis in the liver was suppressed after intranasal administration of GALP. Lipid metabolome analysis of the livers showed an increase in the factors involved in feeding inhibition, a possible increase in fatty acid oxidation, and an improvement in fatty acid composition. Taken together, these findings suggest that the anti-obesity effects of GALP may occur through feeding inhibition as well as through improvement of hepatic lipid metabolism. These results suggest that intranasal administration of GALP may be an effective means of obesity prevention and treatment. Our research group intends to further analyze the detailed mechanism of the anti-obesity effect of intranasal administration of GALP and conduct experiments for future clinical application to humans.

## Figures and Tables

**Figure 1 ijms-24-15825-f001:**
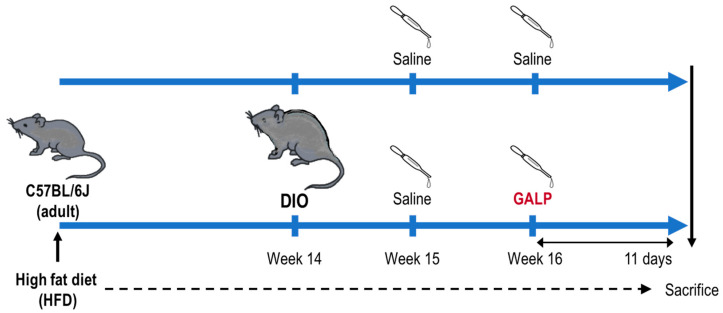
Experimental design and schedule of intranasal galanin-like peptide (GALP) for investigating changes in body weight of diet-induced obese (DIO) mice.

**Figure 2 ijms-24-15825-f002:**
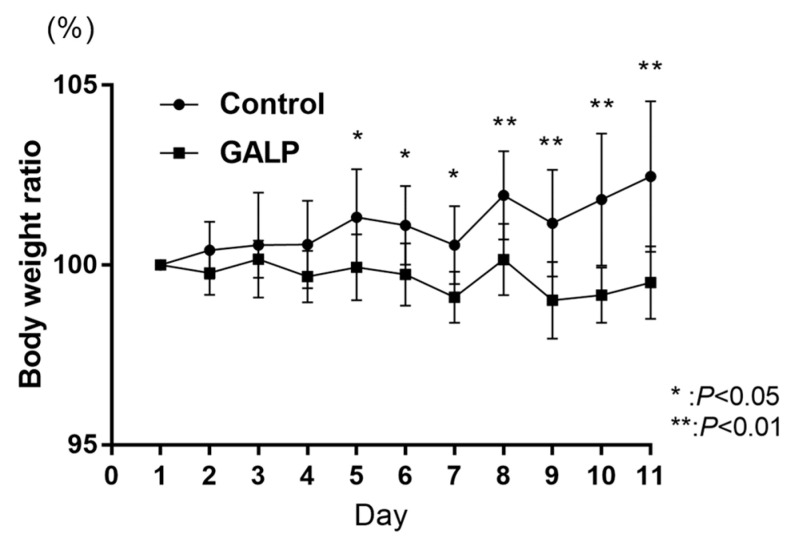
Change in body weight of diet-induced obese (DIO) mice receiving high-fat diet and intranasal saline or galanin-like peptide (GALP). Body weight on day 1 of treatment is shown as 100%, and *p*-values are for control vs. GALP treatment.

**Figure 3 ijms-24-15825-f003:**
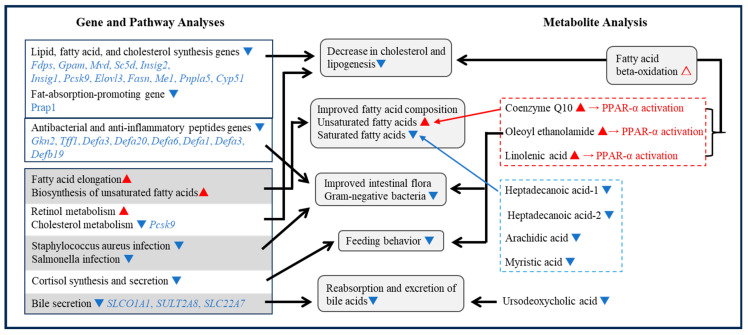
Molecular factors involved in the anti-obesity effect of GALP in the livers of DIO mice. ▲ (Red) = increased, ▼ (Blue) = decreased, and △ (Red) = estimated increase.

**Table 1 ijms-24-15825-t001:** Genes whose expression (fold-change) was altered most in the livers of diet-induced obese (DIO) mice by intranasal administration of GALP. (left) Top 20 genes with increased expression. (right) Top 20 genes with decreased expression.

Gene Name	Fold-Change	Gene Name	Fold-Change
*Fmo3*	10.94	*Gkn2*	0.04
*Wfdc3*	3.36	*Tff1*	0.08
*Lyve1*	3.09	*Defa3*	0.13
*Mt2*	3.05	*Hsd3b4*	0.17
*Olfr1342*	2.94	*Hsd3b5*	0.17
*Aldh1a3*	2.72	*Defa20*	0.18
*Mt1*	2.59	*Clca1*	0.23
*Mt1*	2.58	*Defa6*	0.24
*Slc22a26*	2.57	*Moxd1*	0.25
*Mt1*	2.55	*Defa1*	0.27
*Mt1*	2.55	*Cyp26a1*	0.27
*Mt1*	2.52	*Pnpla5*	0.28
*Mt1*	2.51	*Pdia6*	0.30
*Mt1*	2.50	*Slco1a1*	0.31
*Mt1*	2.49	*Prap1*	0.31
*Mt1*	2.48	*Fdps*	0.32
*Lrtm2*	2.48	*Pik3c2g*	0.38
*Mt1*	2.46	*Defb19*	0.39
*Fgfr1*	2.39	*Krt20*	0.39
*Krt23*	2.37	*Hsd3b1*	0.40

**Table 2 ijms-24-15825-t002:** Metabolites increased (>1.5-fold difference) or reduced (<0.75-fold difference) in the livers of diet-induced obese (DIO) mice treated with intranasal GALP based on lipid metabolomics analysis. ID consists of the initial letter and serial number of the measurement mode, P indicates positive mode, and N indicates negative mode. Metabolites classified as monounsaturated (polyunsaturated) fatty acids, saturated fatty acids, and fatty acids are highlighted in cream, dark gray, respectively. Endocannabinoids are shown in blue letters and bile acids in green letters. Red, increased; Blue, decreased.

ID	Compound Name	Relative Area		Comparative Analysis
		Vehicle	GALP	Ratio
				GALP vs Vehicle
P_0057	Coenzyme Q10	N.D.	0.0001	1<
P_0043	Hydroxyprogesterone caproate	N.D.	0.0001	1<
P_0010	Oleoyl ethanolamide AEA(18:1)	0.0002	0.0006	3.38
P_0052	1,2-Dipalmitoyl-glycero-3-phosphoethanolamine	0.0003	0.0006	2.22
P_0023	AC(14:0)-1	0.0001	0.0001	2.18
N_0031	Erucic acid	0.0000	0.0001	1.99
N_0037	Deoxycholic acid	0.0000	0.0000	1.84
P_0028	5α-Cholestan-3-one-2	0.0001	0.0002	1.72
N_0047	Taurocholic acid	0.0350	0.0596	1.70
N_0046	Taurochenodeoxycholic acid	0.0047	0.0078	1.65
N_0012	Linolenic acid	0.0000	0.0000	1.57
P_0042	AC(18:0)-2	0.0002	0.0003	1.52
P_0025	2-Arachidonoylglycerol	0.0013	0.0019	1.51
N_0030	FA(22:3)	0.0001	0.0000	0.74
P_0049	α-Tocopherol acetate	0.0006	0.0004	0.73
N_0011	Heptadecanoic acid-2 FA(17:0)-2	0.0001	0.0001	0.67
P_0038	γ-Tocopherol	0.0003	0.0002	0.67
P_0050	Zeaxanthin	0.0001	0.0000	0.67
N_0023	Arachidic acid	0.0001	0.0000	0.66
P_0006	Palmitoylethanolamide-2	0.0004	0.0003	0.62
P_0048	AC(22:0)	0.0003	0.0002	0.61
P_0016	AC(13:1)	0.0003	0.0002	0.59
N_0003	Myristic acid	0.0001	0.0001	0.59
P_0008	AC(10:0)	0.0019	0.0011	0.56
N_0001	FA(12:0)	0.0001	0.0000	0.50
N_0010	Heptadecanoic acid-1 FA(17:0)-1	0.0001	0.0000	0.48
N_0004	FA(15:0)	0.0000	0.0000	0.45
N_0008	FA(17:1)-1	0.0000	0.0000	0.39
P_0024	Riboflavin	0.0001	0.0000	0.23
P_0047	AC(20:0)-1	0.0001	N.D.	<1
N_0009	FA(17:1)-2	0.0000	N.D.	<1
P_0001	Flavanone	0.0004	N.D.	<1
N_0038	Ursodeoxycholic acid	0.0000	N.D.	<1
	Mono (Poly) unsaturated fatty acid			
	Saturated fatty acid			
Endocannabinoids				
Bile acid				

FA indicates fatty acid, AC indicates acylcarnitine, and the numbers in parentheses indicate the number of carbons and double bonds. ID consists of the initial letter and serial number of the measurement mode, where P indicates positive mode and N indicates negative mode. N.D.: Not Detected. Analyzed, but below detection limit.

**Table 3 ijms-24-15825-t003:** Results of KEGG pathway analysis of the genes whose expression levels were altered in the livers of DIO mice by intranasal administration of GALP. The pathways to be described further in the Discussion section are shown in red.

**Category KEGG_PATHWAY**		
**UP**		
mmu00062	Fatty acid elongation	ACOT1, ELOVL7, ACOT3
mmu01040	Biosynthesis of unsaturated fatty acids	ACOT1, ELOVL7, ACOT3
mmu00140	Steroid hormone biosynthesis	CYP2B13, SULT1E1, CYP2B10
mmu00830	Retinol metabolism	ALDH1A3, CYP2B13, CYP2B10
mmu04010	MAPK signaling pathway	GADD45B, MAP3K6, FGFR1
mmu04978	Mineral absorption	MT2, MT1
mmu04913	Ovarian steroidogenesis	ACOT1, ACOT3
mmu05218	Melanoma	GADD45B, FGFR1
**Down**		
mmu05150	* Staphylococcus aureus * infection	CFD, DEFA6, DEFA3, DEFA1, DEFA20, KRT20
mmu04927	Cortisol synthesis and secretion	NR4A1, HSD3B4, HSD3B5, STAR, HSD3B1
mmu04925	Aldosterone synthesis and secretion	NR4A1, HSD3B4, HSD3B5, STAR, HSD3B1
mmu04913	Ovarian steroidogenesis	HSD3B4, HSD3B5, STAR, HSD3B1
mmu04934	Cushing syndrome	NR4A1, HSD3B4, HSD3B5, STAR, HSD3B1
mmu00900	Terpenoid backbone biosynthesis	FDPS, IDI1, MVD
mmu00140	Steroid hormone biosynthesis	HSD3B4, HSD3B5, CYP2C66, HSD3B1
mmu04915	Estrogen signaling pathway	TFF1, FOS, KRT20, HSPA1A
mmu04979	Cholesterol metabolism	STAR, SORT1, PCSK9
mmu04621	NOD-like receptor signaling pathway	DEFA6, DEFA3, DEFA1, DEFA20
mmu04024	cAMP signaling pathway	HCN4, SUCNR1, PDE4B, FOS
mmu05202	Transcriptional misregulation in cancer	DEFA6, DEFA3, DEFA1, DEFA20
mmu04976	Bile secretion	SLCO1A1, SULT2A8, SLC22A7
mmu05132	* Salmonella * infection	MYO6, PIK3C2G, FOS, TUBA4A
mmu00100	Steroid biosynthesis	SC5D, CYP51
mmu04010	MAPK signaling pathway	EFNA1, NR4A1, FOS, HSPA1A

## Data Availability

The data presented in this study are available in the article and submitted databases (GEO). The raw data are available upon a reasonable request from the corresponding author.

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
