# Peer review of "Transcriptomic (DNA Microarray) and Metabolome (LC-TOF-MS) Analyses of the Liver in High-Fat Diet Mice after Intranasal Administration of GALP (Galanin-like Peptide)"

_ijms, 2023, doi:10.3390/ijms242115825_

Round 1
Reviewer 1 Report
Comments and Suggestions for Authors
This is an interesting study that investigated efficacy of intranasal administration of Galanin-like Peptide (GALP). The authors used DNA microarray and metabolomics to identify differentially expressed genes and metabolites in the liver tissue of mice switched from HFD to intranasal GALP. The results indicate that intranasal administration of GALP resulted in significant body weight reduction. In addition, liver DNA microarray analysis identified several upregulated and downregulated genes. Liver metabolite analysis also identified several upregulated and downregulated metabolites in different biological pathways. The study is interesting, the manuscript is well-written, and results are clearly described. There are few comments
1- Table 1 change “fold” to “fold change”.
2- Please also add the p value or adjusted p value.
3- Why did the authors use DNA microarray but not RNA sequencing?
Author Response
Comments and Suggestions for Authors
This is an interesting study that investigated efficacy of intranasal administration of Galanin-like Peptide (GALP). The authors used DNA microarray and metabolomics to identify differentially expressed genes and metabolites in the liver tissue of mice switched from HFD to intranasal GALP. The results indicate that intranasal administration of GALP resulted in significant body weight reduction. In addition, liver DNA microarray analysis identified several upregulated and downregulated genes. Liver metabolite analysis also identified several upregulated and downregulated metabolites in different biological pathways. The study is interesting, the manuscript is well-written, and results are clearly described. There are few comments
ANSWER= Thank you very much for the time to review our manuscript and the comments; in consideration of your and the 2nd reviewer comments, the manuscript has been revised thoroughly; and the English Editing by a Native Speaker has been done for clarity in writing and expression. All CORRECTIONS/EDITS are marked in RED in the revised manuscript.
- Table 1 change “fold” to “fold change”.
ANSWER= yes, corrected, to fold-change (Revised Table 1).
- Please also add the p value or adjusted p value.
ANSWER= yes, Figure 1 was modified into a line chart as Figure 2, and the p value has been added.
- Why did the authors use DNA microarray but not RNA sequencing?
ANSWER= thank you for this question, indeed we have been doing DNA microarrays since 2000, and have standardized the technique across various organisms, which gives us greater confidence in its usage and application for ‘monitoring gene activity via mRNA abundance across all transcripts’. Moreover we have from the very beginning used the ‘dye-swap’ technique to obtain highly significant data, and also a method to confirm the best quality of total RNA. In other words, DNA microarray is well validated, and researchers are familiar with this technique and the obtained data sets. It is also a good comparative tool, and can be used to compare large number of samples with relative ease and quickly provide differences in gene expressions, though it is dependent on the probe design. It is also less expensive (more so now than at the beginning) than the NGS, which is yet to be widely used – i.e., as widely used as the DNA microarrays. Of-course, the NGS has its advantages, ‘no prior data required for the experiment’, ‘high discovery potential’, and no hardware redesign’.; but saying that it is still not easy to identify/validate gene expression changes with ease as compared to the DNA microarray. Therefore, at present, with our vast experience and highly standardized procedure for sample preparation, total RNA extraction and quality, we would prefer to use this technique for our numerous experiments with a large number of samples for comparative analyses.
Reviewer 2 Report
Comments and Suggestions for Authors
Takenoya et al investigated the efficacy of intranasal administration of galanin-like peptide (GALP) in preventing obesity in mice fed a high-fat diet (HFD). The researchers used an omics-based approach and LCMS to identify the genes and metabolites that are changed by GALP administration and to determine the mechanism of action of GALP in reducing body weight. The study suffers from inappropriate experimental design, figures, data analysis, and written structure which prevents its acceptance for publication.
How do authors conclude that GALP administration reduces body weight ratio if the diet has been modified during the experiment? Diet modification is implied in the manuscript by the word “switched” -- (“… liver tissue of mice switched from HFD to intranasal GALP …”). The methods section does not describe how the mice were fed after 2 weeks of acclimatization. The authors mention that GALP is effective in suppressing food intake, but the mice were not given the same HFD as before. The mice should have been offered the same diet in order to understand the GALP-induced food intake suppression. Maybe the GALP administration was not going to suppress HFD intake and we cannot know that.
Figure 2 is not useful and should be removed. There is no point in this figure, there are just two numbers 53 and 165. Authors should properly depict the results with a scatter plot with error bars instead of this figure so that the readers can have an idea of what the certainties of the fold-changes in each gene expression are. That certainty could inform the authors whether or not they should focus on a specific gene.
What is the rationale for setting the weight loss threshold for the stratification of mice into two groups (with less and greater weight loss)? Fig 1 shows 8 data points that are all very close to each other. Actually, all but one each from each tail of the distribution is within the standard deviation of the mean (assuming the whiskers show the SD, authors should make this clear). Authors need to come up with a scientific rationale for this decision or remove the parts related to the grouping.
“This result of significant weight loss by intranasal GALP administration was similar to the result of a previous experiment [3] in which GALP was administered intranasally once daily for 7 days, suggesting that GALP is indeed effective in reducing body weight.” This is very confusing, I see that the reference [3] is a self-citation to the authors’ 2011 publication and so the authors mention here that they administered GALP once daily for 7 days before but how is this supportive of the hypothesis that GALP is effective in reducing body weight now that they do once daily for 11 days?
Even if the experimental design has been corrected these results would not be able to prove the hypothesis, but the authors say -direct quote- “These experimental results suggest that intranasal administration of GALP is an extremely effective means of obesity prevention and treatment.” These are overstatements and authors should rewrite the conclusions so that it is not sweeping as it is now but it is more prudent.
I checked reference [3] and saw that the cited results have not been shown there -direct quote- “Intranasal infusion of GALP significantly reduced body weight over the course of a week. However, food and water intake, as well as locomotor activity, remained unchanged (Shiba et al., manuscript in preparation).” Can the authors correct this inappropriate self-citation?
A lot of unnecessary details under section 2. Sections 2.2, 2.3, and most of 2.4 cannot be considered as results. Only the necessary parts of this sort of literature review should be mentioned under the discussions section as they become relevant.
The authors should have used female mice as well. Obesity affects both genders nearly equally.
Would the authors consider investigating the effects of GALP on mice on a high-sugar diet?
Fig 1, description of the asterisks, whiskers and the statistical test used are missing.
Authors should also publish the raw data for LCMS. As a supplementary for example.
Why is there a threshold for fold change for increased but not decreased gene expression?
Comments on the Quality of English LanguageLanguage needs extensive editing for clarity and grammar. Currently, the manuscript is incomprehensible since it lacks clarity in its claims, abstract, introduction, methods, and discussions. Below are a few examples of many such occurrences.
“…using an omics-based approach, this study identified several variable genes and metabolites in the liver tissue of mice switched from HFD to intranasal GALP by whole genome DNA microarray” repetition and clarity
“The effects of intranasal GALP administration in alleviating weight gain in male mice fed HFD included obesity, antioxidant, anti-inflammatory, and fatty acid metabolism involved in metabolites and genetic alterations” unclear, for example, obesity is not an effect, metabolism is not an effect, what does “metabolism involved in metabolites” mean?
“…by intranasal GALP administration, as ARE lipid and cholesterol synthesis, fat absorption and bile uptake” grammar
fig 2 caption: “Down: The number of genes whose expression level increased to 0.75-fold by intranasal administration of GALP was 53.” inconsistent. Is it increased or decreased?
“GALP treatment increased the expression of genes that were 1.5-fold or 0.75-fold higher than that of the control in both the control and GALP-treated (group with greater weight loss) and control and GALP-treated (group with less weight loss) groups.”Unclear. Also, mention the N here for groups with less and greater weight loss please.
“2.2.1 Genes who’s Expression was Increased by GALP Administration” grammar
Author Response
Takenoya et al investigated the efficacy of intranasal administration of galanin-like peptide (GALP) in preventing obesity in mice fed a high-fat diet (HFD). The researchers used an omics-based approach and LCMS to identify the genes and metabolites that are changed by GALP administration and to determine the mechanism of action of GALP in reducing body weight. The study suffers from inappropriate experimental design, figures, data analysis, and written structure which prevents its acceptance for publication.
ANSWER= Thank you very much for the time to review our manuscript and the critical comments-these are much appreciated as they help use re-read the manuscript while adding the parts/missing texts/explanations and errors as pointed out by the expert referee. Specifically, the referee mentions problems with the inappropriate experimental design, figures, data analysis, and text structure, and we appreciate the referee’s opinions, and we will respond to the best and in good faith and request consideration of our responses to your comments in this ANWERS letter, and the REVISED manuscript. Thank you.
And, in consideration of your and the 1st reviewer comments, the manuscript has been revised thoroughly for clarity and to answer each comment raised. We have also added some points missed out in our original submission. The English language editing by a Native Speaker has been done for clarity in writing and expression. All CORRECTIONS/EDITS are marked in RED in the revised manuscript.
How do authors conclude that GALP administration reduces body weight ratio if the diet has been modified during the experiment? Diet modification is implied in the manuscript by the word “switched” -- (“… liver tissue of mice switched from HFD to intranasal GALP …”). The methods section does not describe how the mice were fed after 2 weeks of acclimatization. The authors mention that GALP is effective in suppressing food intake, but the mice were not given the same HFD as before. The mice should have been offered the same diet in order to understand the GALP-induced food intake suppression. Maybe the GALP administration was not going to suppress HFD intake and we cannot know that.
ANSWER= Due to a lack of explanation of the experimental methodology, the experimental design / schedule is presented in a new Figure 1. In this study/experiment, animals were first fed a high-fat diet for 14 weeks to create highly obese DIO mice. Then, GALP and saline control were administered intranasally to see if GALP could suppress the hyperobesity state of the animals and prevent weight gain. This experiment was conducted to determine if intranasal administration of GALP had an inhibitory effect on weight gain compared to the control (saline), while the animals were consistently fed a high-fat diet. Previous experiments have been conducted using the same experimental design as the present study using ob/ob and DIO mouse models of type 2 diabetes and obesity, and the anti-obesity effects of GALP administration have been similarly observed. All diets used in this experiment were high-fat diets, and we apologize for any misunderstanding caused by insufficient explanation in the original submission. "Switched" was not an appropriate term to describe the addition of intranasal GALP to HFD and the experimental design has now been clarified.
Figure 2 is not useful and should be removed. There is no point in this figure, there are just two numbers 53 and 165. Authors should properly depict the results with a scatter plot with error bars instead of this figure so that the readers can have an idea of what the certainties of the fold-changes in each gene expression are. That certainty could inform the authors whether or not they should focus on a specific gene.
ANSWER= Thank you for correcting us, and yes, it does not make much sense here, and thus we have decided to delete the Figure 2, and revised Figure 1 is presented as Figure 2 now.
What is the rationale for setting the weight loss threshold for the stratification of mice into two groups (with less and greater weight loss)? Fig 1 shows 8 data points that are all very close to each other. Actually, all but one each from each tail of the distribution is within the standard deviation of the mean (assuming the whiskers show the SD, authors should make this clear). Authors need to come up with a scientific rationale for this decision or remove the parts related to the grouping.
ANSWER= Initially, mice were divided into two groups (a group with less suppression of weight gain and a group with a high suppression of weight gain) and DNA microarray analysis was performed to identify genes involved in the weight gain suppression in order to determine if there were differences or characteristics in gene expression changes between groups with little or high suppression of weight gain. This was thought to be useful for identification of gene expression changes at the whole genome level. However, since there was no significant difference, a list of genes whose expression changed in common was compiled from the DNA microarray analysis results of the two groups (less suppression of weight gain group and high suppression of weight gain group) and used for the subsequent analysis.
We apologize for a lack of an explanation for the above, and we describe it as follows: Initially, in the process of identifying genes involved in GALP-induced weight loss in the microarray analysis, we thought it would be effective to group the genes according to the degree of weight loss. However, as there was no significant difference between the results of each group, we decided to proceed with the analysis by selecting genes that showed common expression changes in each group, as shown below.
“This result of significant weight loss by intranasal GALP administration was similar to the result of a previous experiment [3] in which GALP was administered intranasally once daily for 7 days, suggesting that GALP is indeed effective in reducing body weight.” This is very confusing, I see that the reference [3] is a self-citation to the authors’ 2011 publication and so the authors mention here that they administered GALP once daily for 7 days before but how is this supportive of the hypothesis that GALP is effective in reducing body weight now that they do once daily for 11 days?
Even if the experimental design has been corrected these results would not be able to prove the hypothesis, but the authors say -direct quote- “These experimental results suggest that intranasal administration of GALP is an extremely effective means of obesity prevention and treatment.” These are overstatements and authors should rewrite the conclusions so that it is not sweeping as it is now but it is more prudent.
ANSWER= Yes, the referee is correct in pointing out this error from our side; we apologize for it. First of all, the reference will be corrected to [4], not [3]. As the referee pointed out, this result of significant weight loss by intranasal GALP administration is similar to a previous experiment [4] in which GALP was administered intranasally once daily for 7 days to diet-induced obese (DIO) mice, suggesting that GALP is indeed effective in weight loss, but it was overstated. The results of this study are similar to the results of a previous experiment [4]. As you pointed out, since the number of days of administration differs between this experiment and the previous experiment, we would like to revise our definitive discussion of GALP's "inhibition of weight gain" in conjunction with our previous paper [4] in the revised text/manuscript.
I checked reference [3] and saw that the cited results have not been shown there -direct quote- “Intranasal infusion of GALP significantly reduced body weight over the course of a week. However, food and water intake, as well as locomotor activity, remained unchanged (Shiba et al., manuscript in preparation).” Can the authors correct this inappropriate self-citation?
ANSWER= THANK YOU for your critical comment again-it has helped us correct our error. As mentioned above, the citation in reference [3] was incorrect. It will be corrected in reference [4]. In the experiment of [4], once-daily intranasal administration of GALP significantly reduced the body weight of diet-induced obese (DIO) mice by day 5. Intranasally administered GALP reduced the cumulative body weight of DIO mice despite reduced locomotor activity. ob/ob mice showed reduced food intake, water intake, weight gain, and locomotor activity without taste aversion. Similar effects were observed in DIO mice.
A lot of unnecessary details under section 2. Sections 2.2, 2.3, and most of 2.4 cannot be considered as results. Only the necessary parts of this sort of literature review should be mentioned under the discussions section as they become relevant.
ANSWER= Yes, some aspects of the discussion on the genes is present in these sections; and will be deleted from the results, and rewritten/moved to the discussion section in an appropriate form. See Discussion section, in RED.
The authors should have used female mice as well. Obesity affects both genders nearly equally.
ANSWER= Thank you for the advice, and as of now, we have conducted research mainly on male animals. We believe that experiments with female mice would be of great interest, and we hope to conduct anti-obesity experiments with female mice in the future.
Previous reports have included experiments that examined feeding behavior in females after GALP administration. There is a 24-hour weight loss with female ICV administration, although it is not nasal administration (https://pubmed.ncbi.nlm.nih.gov/16042964/).
Would the authors consider investigating the effects of GALP on mice on a high-sugar diet?
ANSWER= At present, the physiological effects of GALP are focused on new clinical research on weight loss, and there are no plans to conduct experiments on the effects of GALP on glucose metabolism. And, yes, in relation to the body weight, alarin, a regulatory peptide on the galanin family, may facilitate insulin sensitivity and glucose uptake through the central alarin projective system. If there is an opportunity to conduct experiments in the future, we would like to do so. Thank you for your suggestion.
Fig 1, description of the asterisks, whiskers and the statistical test used are missing.
ANSWER= Figure 1 was modified into a line chart as Figure 2 and the P-value was included.
Authors should also publish the raw data for LCMS. As a supplementary for example.
ANSWER= The data has been presented as a new Supplementary Table 1.
Why is there a threshold for fold change for increased but not decreased gene expression?
ANSWER= Thank you for your comment, and apologies for not being able to understand your question and your meaning of threshold. The gene expression change by the DNA microarray analysis in this experiment is used in the analysis as 1.5 times or more than the value calculated by Fold Change=GALP/Control as increased expression and 0.75 times or less as decreased expression of the gene.
Comments on the Quality of English Language
Language needs extensive editing for clarity and grammar. Currently, the manuscript is incomprehensible since it lacks clarity in its claims, abstract, introduction, methods, and discussions. Below are a few examples of many such occurrences.
ANSWER= With the chance to revise the manuscript – we have re-read the manuscript thoroughly and with each comment answered, revised it together with an improvement in the English language, we have also used a professional Native English speaker/researcher to help improve the clarity of the text of the revised manuscript.
“…using an omics-based approach, this study identified several variable genes and metabolites in the liver tissue of mice switched from HFD to intranasal GALP by whole genome DNA microarray” repetition and clarity
ANSWER= This has been corrected/re-phrased.
“The effects of intranasal GALP administration in alleviating weight gain in male mice fed HFD included obesity, antioxidant, anti-inflammatory, and fatty acid metabolism involved in metabolites and genetic alterations” unclear, for example, obesity is not an effect, metabolism is not an effect, what does “metabolism involved in metabolites” mean?
ANSWER= This has been corrected/re-phrased.
“…by intranasal GALP administration, as ARE lipid and cholesterol synthesis, fat absorption and bile uptake” grammar
fig 2 caption: “Down: The number of genes whose expression level increased to 0.75-fold by intranasal administration of GALP was 53.” inconsistent. Is it increased or decreased?
ANSWER= This has been corrected/re-phrased.
“GALP treatment increased the expression of genes that were 1.5-fold or 0.75-fold higher than that of the control in both the control and GALP-treated (group with greater weight loss) and control and GALP-treated (group with less weight loss) groups.”Unclear. Also, mention the N here for groups with less and greater weight loss please.
ANSWER= This has been corrected/re-phrased.
“2.2.1 Genes who’s Expression was Increased by GALP Administration” grammar
ANSWER= This has been corrected/re-phrased.
THANK YOU ONCE AGAIN to the Referee 2, for the critical reading and comments, which has helped us improve the manuscript.